# Computational Assessment of Magnetic Nanoparticle Targeting Efficiency in a Simplified Circle of Willis Arterial Model

**DOI:** 10.3390/ijms24032545

**Published:** 2023-01-29

**Authors:** Rodward L. Hewlin, Joseph M. Tindall

**Affiliations:** 1Center for Biomedical Engineering and Science (CBES), Department of Engineering Technology, University of North Carolina at Charlotte, Charlotte, NC 28223, USA; 2Applied Energy and Electromechanical Systems (AEES), Department of Engineering Technology, University of North Carolina at Charlotte, Charlotte, NC 28223, USA

**Keywords:** cardiovascular disease, capture efficiency, cerebral arterial network, computational model, Eulerian–Lagrangian, magnetizable nanoparticles, medical drug targeting

## Abstract

This paper presents the methodology and computational results of simulated medical drug targeting *(MDT)* via induced magnetism intended for administering intravenous patient-specific doses of therapeutic agents in a Circle of Willis (CoW) model. The multi-physics computational model used in this work is from our previous works. The computational model is used to analyze pulsatile blood flow, particle motion, and particle capture efficiency in a magnetized region using the magnetic properties of magnetite *(Fe_3_O_4_)* and equations describing the magnetic forces acting on particles produced by an external cylindrical electromagnetic coil. A Eulerian–Lagrangian technique is implemented to resolve the hemodynamic flow and the motion of particles under the influence of a range of magnetic field strengths *(B_r_ = 2T, 4T, 6T, and 8T)*. Particle diameter sizes of 10 nm to 4 µm in diameter were assessed. Two dimensionless numbers are also investigated *a priori* in this study to characterize relative effects of Brownian motion *(BM)*, magnetic force-induced particle motion, and convective blood flow on particle motion. Similar to our previous works, the computational simulations demonstrate that the greatest particle capture efficiency results for particle diameters within the micron range, specifically in regions where flow separation and vortices are at a minimum. Additionally, it was observed that the capture efficiency of particles decreases substantially with smaller particle diameters, especially in the superparamagnetic regime. The highest capture efficiency observed for superparamagnetic particles was 99% with an 8T magnetic field strength and 95% with a 2T magnetic field strength when analyzing 100 nm particles. For 10 nm particles and an 8T magnetic field strength, the particle capture efficiency was 48%, and for a 2T magnetic field strength the particle capture efficiency was 33%. Furthermore, it was found that larger magnetic field strengths, large particle diameter sizes *(1 µm and above),* and slower blood flow velocity increase the particle capture efficiency. The key finding in this work is that favorable capture efficiencies for superparamagnetic particles were observed in the CoW model for weak fields *(B_r_ < 4T)* which demonstrates MDT as a possible viable treatment candidate for cardiovascular disease.

## 1. Introduction

Cardiovascular disease remains the leading cause of hospitalization and death worldwide [1,2,3,4,5]. Medical drug delivery via the use of magnetizable nanoparticles as carrier vehicles for therapeutic agents under the influence of a magnetic field in cardiovascular flow has received much attention for the prospects of treatment processes for both cancer and cardiovascular disease [6,7,8,9]. This drug delivery scheme promises improvements in diagnosis and therapeutic treatments by increasing targeting efficacy while avoiding and/or minimizing systemic circulation which often causes added toxicity to healthy sites. Medical drug targeting *(MDT)* refers to the attachment of therapeutic agents to magnetizable nanoparticles to concentrate them at the desired target location under an applied magnetic field. This method requires extensive investigation of the external field strength and its interaction with carrier particles. 

The substantial difficulties experienced and reported in the literature with MDT include the inherently weak magnetic forces produced by the magnetic field relative to the hydrodynamic forces produced by cardiovascular flow in targeting regions [10,11,12]. This makes MDT challenging, provided that the magnetic force on a carrier particle is proportional to not only the magnitude of the magnetic field, but also to the gradient. Furthermore, other properties and parameters influence MDT such as carrier particle material and diameter, magnetic field strength, magnet distance from the target region, and flow parameters such as hydrodynamic force, wall shear stress, and vorticity [13]. Various experimental studies have been reported on characterizing the behavior of magnetic nanoparticles in research and clinical applications for both cancer and cardiovascular disease. 

Advances in this area of research have facilitated the production of micro- and nanostructures with great precision [14]. In addition to spherical particle carrier vehicles used in early experiments, state-of-the-art drug delivery systems incorporate parcels of nanotubes to sheath biochemically active components. These carrier structures/vehicles can be designed for various specifications [15,16] while a viable compromise between competing requirements should be investigated. For example, larger magnetic particles with micrometer radii are easier to manipulate via external fields, as the forces acting on them are proportional to their volume. Conversely, the use of superparamagnetic iron oxide particles *(SPIONS)* can substantially increase magnetic susceptibility, and hence enhance the response of particles to an external magnetic field [17,18]. 

There has been a growing interest in the scientific and clinical application of magnetic nanoparticles as MDT vehicles for the development of efficient treatment strategies. SPIONS have also been used as magnetic resonance imaging *(MRI)* contrast agents for labelling mammalian cells since their features can be easily tailored to include targeting moieties, fluorescence dyes, or therapeutic agents. SPIONS can also be taken up by the cells through endocytosis and one particular SPION that contains ferumoxides is approved for hepatic imaging by the US Food and Drug Administration *(FDA)* [19]. There is significant interest in using MDT for the treatment of diseases such as cancer [20] due to the need to maximize damage to tumor cells while keeping the exposure to healthy tissue in the remainder of a patient’s body within tolerable levels. There have been a number of preclinical studies [21,22,23]—including a phase I clinical human trial using a single permanent magnet to concentrate epidoxorubicin-coated magnetic nanoparticles within shallow, inoperable tumors—but with a number of issues identified [24].

One of the key goals of MDT is to reach targets *(e.g., tumors and plaque buildup)* deeper within the body and arterial network, but different locations can require very different magnetic nanoparticle properties and magnetic field strength settings. In vitro experiments with flow phantoms can be used to determine the behavior of magnetic nanoparticles with different physiological and physical parameters [25,26]. Simulation work by Nacev et al. suggests the use of a feedback control algorithm that modifies the applied magnetic field based on accurate real-time information on the distribution of particles to focus the particles at a particular site [27]. To be a predictor tool in bioresearch, MDT simulations must include a range of physical phenomena. Furthermore, to be able to resolve processes on relevant time and length scales, the simulation tools used must be computationally efficient. An ideal model would account for the mechanical properties of vessel walls, the complex rheological behavior of blood and its particulate nature, external magnetic fields, and gravity, etc. However, careful evaluation and control of the errors arising from different modeling assumptions and simplifications should enable reduced *(and computationally efficient)* models to be used with accuracy and reliability in clinical decision support. Moreover, multiscale models can inform coarse grained parametrization by quantifying effective parameter values.

Despite ongoing research of cardiovascular flows and MDT as a potential candidate for medical drug delivery for cancer and cardiovascular disease, the fluid dynamics as well as the magnetic effects associated with MDT need further investigation. MDT methods have not yet been approved nor have they reached the stage of clinical use. This paper presents simulated medical drug targeting *(MDT)* via induced magnetism intended for administering intravenous patient-specific doses of therapeutic agents in the Circle of Willis. The main contributions of the proposed work include:Theoretical methodology for computational modelling of simulated MDT in the Circle of Willis *(CoW)* model.Results of blood flow hydrodynamics and its potential effects on MDT.MDT results that show favorable capture efficiencies for micron range particles and a potential for enhancing capture efficiency of superparamagnetic particles in small *(diameter < 32 mm)* arteries.

All the simulations are performed using ANSYS Fluent^TM^ 22. The blood flow in the arterial vessel is modelled as non-Newtonian, incompressible, laminar, and transient. The forces that have effects on the particles are viscous drag, buoyancy, and magnetic forces. A cylindrical electromagnetic coil is modelled as the source of the external magnetic field. The effects of the external magnetic field, particle size, and hydrodynamic effects on capturing particles are discussed. Two dimensionless numbers are introduced to characterize effects of these factors on particle transport. The next section presents the methodology of this work.

## 2. Results

This section of the paper presents the computational results of this work. To effectively target particles in the cardiovascular system, the magnetic force must overcome the hydrodynamic drag force produced by cardiovascular flow. Cardiovascular flow is highly unsteady and has a vast effect on hydrodynamic parameters. The flow characteristic profile inside the CoW arterial model is shown in Figure 1 for systole during cycle 5 of the cardiac flow waveform. 

Flow streamlines were plotted to represent the trajectory of finite fluid particles that are tracked and superimposed on the transparent CoW mesh as shown in Figure 1. As depicted in Figure 1, at systole, a Womersley flow profile is established at the bifurcation region of the basilar, left, and right carotid bifurcation arteries and continues throughout the arterial network for the cardiac cycle observed. In the middle cerebral artery regions, the flow is highly skewed toward the wall as shown in the axial velocity contours. The streamlines are also disorganized in the bifurcating regions of the basilar, left, and right carotid artery and show areas of flow separations and secondary flow. Similar findings have also been reported, whereby skewed flow structure is primarily caused by the misalignment of the mean axis of the artery, curvature of the lumen, unsteady centrifugal forces caused by the sudden decrease and increase in flow velocity [28,29,30]. 

The asymmetric flow pattern within the CoW model plainly depicts strong unsteadiness *(not to be confused with turbulence as the Re = 658 for systole)* in middle cerebral arteries, while remaining flow dampens downstream to the anterior cerebral artery region and bifurcating regions. This is believed to be due to the large momentum increase of fluid motion from systole to diastole in which the flow has achieved maximum flow. The Womersley profiles also indicate possible recirculating flow in these regions. A more conclusive examination can be predicted from a plot of the velocity contours and profile in the middle cerebral artery shown in Figure 2 and Figure 3. Figure 2 shows the velocity magnitude contours superimposed with the velocity magnitude vectors at systole and diastole in a slice of the middle cerebral artery *(artery closest to the magnet as shown in the schematic).* A counterclockwise rotating vortex as depicted by the velocity vectors is also present in the flow. The Womersley profile in the middle cerebral artery is verified by the velocity profiles plotted in Figure 3. 

Owing to inertia, flow inside branched vessels of the arterial system tend to be pushed toward one side of the branch vessel. This effect is apparent in the in-plane velocity vectors shown in Figure 2. There is a counter rotating vortex formed due to the pulsatile flow and inertia depicted in Figure 2. In Figure 1, from the bifurcation region of the basilar, left, and right carotid bifurcation artery, the flow changes significantly and becomes more chaotic with pronounced vortical coherent structures and strong central vortex threading being formed throughout the curved regions and branches. Depending on the magnitude of the flow, this will make targeting particles difficult in these regions. Vorticity, which describes the magnitude of spin also gives insight into the magnitude of flow recirculation.

Figure 4 provides a contour plot of vorticity at systole for cycle 5. The largest recirculation zones appear in bending and bifurcating branch regions as mentioned previously. These zones are the regions where the pronounced vortical coherent structures and strong central vortex threading are being formed which will make targeting particles difficult. These unsteady affects also effect WSS as shown in Figure 5. 

The average range of WSS in the cerebral artery network is 1–33 Pa [31,32], whereas in the present work the highest WSS occurs around 60 Pa at bending and bifurcating branch regions, indicating high near wall forces. This may be due to areas where the model possesses sharp transitions and connections. Figure 6 shows the pressure contours systole.

The breakdown of vortex structure in regions can also be observed. Flow separating areas that produce chaotic flow present huge barriers for particle targeting and capture. Moreover, the likelihood of particles sticking to the wall and remaining there are highly unlikely without the presence of a magnetic field and during weak magnetic fields *(B_r_ > 1T).* In this case, the magnitude of hydrodynamic drag should be taken into consideration. 

In this work, there are two particle-capture analysis types performed. The first analysis incorporates a random dispersion of particles with diameters ranging from 10 nm to 4 µm *(heterogeneous dispersion)* uniformly injected into the flow through the basilar artery inlet and subjected to the external magnetic field. The second analysis consists of individual cases of injected particles of one diameter type injected *(homogeneous dispersion)* and subjected to the external magnetic field *(individual case studies of one diameter type performed for varying field strengths (2T, 4T, 6T, and 8T).* A pre-analysis was also performed to determine the drag forces that occur on particles of sizes 10 nm to 4 µm. The calculated drag force range is 0.0458–14.5 pN for particle diameters ranging from 10 nm to 4 µm. This indicates that a magnetic force must be higher than 0.0458 pN for 10 nm diameter particles and 14.5 pN for 4 µm diameter particles to overcome the hydrodynamic drag. The diffusion coefficient, modified Peclet, and *β_m_* of the individual diameter analyses are discussed. 

### Particle Flow and Capture Results

Pulsatile blood flow simulations with a periodic inlet velocity boundary condition were performed in which the particles were uniformly injected at the inlet face at time increments of 1/100th of the flow over a total time of 3.2 s. Over a total time of 3.2 s, a heterogenous mixture of 48,872 magnetite particles was uniformly injected into the CoW arterial model from the inlets of the basilar, left, and right carotid bifurcation artery. Figure 7 shows a contour of the magnetic field strength distribution inside the flow domain for a magnetic field strength of 2T. The average magnetic field strength within the ROI site, specifically the sinus bulb and stented region *(upstream and downstream)* is approximately 0.545T. The magnitude of the force field generated by the magnetic field for 10 nm diameter particles dispersed in the flow field is shown in Figure 8. The highest force observed within the ROI is approximately 52.3 × 10^−6^ pN.

For magnetic field studies, particles were injected through the face of the left carotid bifurcation artery inlet over a total time of 0.8 s with the magnet turned off. After 0.8 s, the magnet was turned on and simulations were run over a total time of 5.8 s. Figure 9a shows an interior view of the injected particles inside CoW arterial model with the magnet turned off. The particles are colored by size and are scaled 100 times the actual diameters. Figure 9b also shows different views of the particles dispersed in the CoW artery model at 3.2 s with the magnetic field turned on.

During the time when the magnet is turned off, the particles are injected into the left carotid artery and flow throughout the CoW due to the viscous drag force produced by the driving pressure wave as shown in Figure 9. The particles follow the flow streamlines as their inertia is negligible compared to the viscous drag. As shown in Figure 9a, over a total time of 3.2 s, the majority of particles have exited the artery and only a few particles are found to adhere to the vessel walls due to the no slip condition. Although the simulations were performed over 5.8 s, the particles clear out at 3.2 s. When the magnet is turned on, as shown in Figure 9b, the particles are found to be influenced appreciably by the magnetic force. The particles are shown to be attracted to the top portion of the vessel and skewed towards the edge of the wall *(nearest to the magnet)* where they can exchange medical drugs with the arterial wall. This phenomenon is illustrated in Figure 10. Figure 10 shows the particle distribution at time 3.4 s for different views. 

As shown in Figure 10a,b, a large cluster of particles are attracted to the top edge of the arterial wall in the middle cerebral artery. The cluster of particles is concentrated in the upper region of the left middle cerebral artery near the magnet. 

Figure 11 shows a contour plot of the homogeneous distribution of 10 nm particles in the CoW model captured at 3.2 s. The homogeneous distribution of particles possesses the same trend in capture efficiency as the heterogeneous dispersion. Most of the particles are attracted to the outer wall near the magnet.

For the homogeneous dispersions, the captured particle density distribution along the axial direction is large, which corresponds to the magnitude of the magnetic force and small magnitude of vorticity in that section. Calculations of *β_m_* and *Pe_m_* were performed to evaluate the relative dominant effect of particle motion in each magnetic field strength setting and particle diameter case. Although the calculations are not shown in this work, the calculations revealed that *β_m_* increases *(β_m_ >> *1*)* with increasing magnetic field strength, indicating a more significant impact of magnetic force on particle motion with larger magnetic field strength. More particles appear to be attracted to the outer wall *(near the magnet*) and the capture efficiency increases correspondingly under a higher magnetic field strength. For *Pe_m_*, it was observed to decrease with increasing magnetic field strength, leading to a faster capturing process because the enhanced magnetic force pulls the particles toward the stented region faster. In this case, the capturing effect is dominant in the particle delivery. This is discussed further in detail and is shown in Figure 11 and Figure 12.

As mentioned previously, the flow is axisymmetric in nature and the magnetic arrangement does not show angular or axial symmetry. The majority of particles tend to be attracted to regions where recirculation is low. No particles are captured near the inner edge of the artery walls of the CoW *(near the centroid of the CoW model and to the right of the centroid as shown in Figure 10)*. The inner edge of middle cerebral arteries and bifurcating regions are high recirculation zones and areas where the Venturi effect causes a significant increase in velocity and makes particle capture difficult. However, when considering the capture efficiency in terms of particle diameter, a significant fraction of the captured particles is 20 nm to 4 µm in diameter. This is an improvement over results that have been reported in previous studies. This is also a positive result when considering ex vivo applications for MDT in large arterial sections and capturing superparamagnetic particles. A capture efficiency quantitative analysis was conducted for homogenous dispersions of particle diameters ranging from the superparamagnetic region to 4 µm. Figure 12 shows the simulation result for MDT capture efficiency as a function of particle diameter. The first observation is that the larger diameter particles *(micron sized particles)* have better cumulative capture efficiency and targeting potential compared to smaller diameter particles *(superparamagnetic particles)* as also reported in previous works.

For particles larger than 1 µm in diameter, high capture efficiencies of 95–99% were observed for the full range of magnetic field strengths. For particles with diameters ranging 50 nm and above, capture efficiencies of 93–99% were observed. For smaller particles ranging from the superparamagnetic regime to 100 nm, capture efficiencies of 20–99% were observed. This is an improvement over our previous work in which we observed a capture efficiency range of 0.5–30% for particles ranging in diameter from 20 to 400 nm and was not subjected to a magnetized implant. Capture efficiencies in the superparamagnetic regime have tripled compared to our previous work when considering a CoW model and varying the field strength *(2T–8T).* This may be due to particles becoming trapped in curved and bifurcating regions, the flow is lower in amplitude compared to larger arteries, and the magnetic field forces are dominate compared to the hydrodynamic drag. Although capture efficiencies are maximized for particles larger than 1000 µm, the key finding in this work is that capture efficiency has been maximized for superparamagnetic particles. As reported in previous studies, for in vivo applications and depending on the medical injection location, large particles (*D* > 200 nm) may be quickly eliminated by the reticuloendothelial system of the spleen and liver during continuous circulation in the cardiovascular system. Smaller particles *(10–200 nm)* in diameter have been reported to be optimal for in vivo MDT as they can escape renal clearance, whereas the large particles are eliminated.

## 3. Methodology

### 3.1. Blood Flow

All blood flow and particle tracking analyses were conducted in ANSYS Fluent^TM^ 22. For blood flow simulations, the continuity and three-dimensional Navier–Stokes equations describing the pulsatile flow of blood as an incompressible fluid is implemented and numerically solved in a finite volume formulation:(1)∇⋅u=0
(2)$$\rho \left(\frac{\partial u}{\partial t}+u\left(u\cdot \nabla \right)\right)=-\nabla p+\nabla \cdot \left(\mu \nabla u\right)$$
where *u* is the three-dimensional velocity vector, *ρ* is the density of blood, *p* is pressure, and *µ* is the dynamic viscosity. The rheology of blood is also considered as described by the non-Newtonian Carreau model:(3)$$\mu (\overset{\cdot }{\gamma })=\mu_{\infty }+(\mu_{0}-\mu_{\infty }){\left(1+{\left(\lambda \overset{\cdot }{\gamma }\right)}^{2}\right)}^{\frac{n-1}{2}}$$
where $\overset{\cdot }{\gamma }$ is the strain rate, *µ_∞_ (0.0035 kg/m∙s)* is the infinite viscosity, *µ*_0_
*(0.056 kg/m∙s)* is the zero-shear viscosity, *λ (3.313)* is the time constant, and *n (0.3568)* is the power law index. A simplified Circle of Willis *(CoW)* model was developed in this work and is discussed further in detail in Section 3.2. A piecewise unsteady velocity waveform is implemented at the inlet of the model as shown in Figure 13 and Equation (4). The unsteady inlet velocity is modeled as fully developed as shown in Equation (5). The walls of the arterial vessels within the CoW are modeled as rigid vessels with the no-slip condition applied at the boundaries.
(4)$$\begin{array}{l} \overset{-}{u}(t)=\\ C_{i}\times \left\{ \begin{array}{l} 83.33t^{3}-12.50t^{2}+0.21667t+0.13\\ 7517.12t^{4}-5688.27t^{3}+1529.70t^{2}-169.61t+6.6462\\ -96.923t^{3}+100.30t^{2}-34.425t+4.1629\\ 58.291t^{2}-5235.7t+1193\\ 18.835t^{3}-36.960t^{2}+23.427t-4.60 \end{array}\right. \end{array}$$
where *C_i_* is the scaling constant for each arterial vessel *(C_i_ = 1 for the basilar artery, C_i_ = 1.4 for the left internal carotid artery, and C_i_ = 1.5 for the right internal carotid artery).*
$$\begin{array}{l} 0s\leq t\leq 0.15s\\ 0.15s\leq t\leq 0.24s\\ 0.24s\leq t\leq 0.42s\\ 0.42s\leq t\leq 0.47s\\ 0.47s\leq t<0.80s \end{array}$$
(5)$$u(y,t)=2\bar{u}(t)\left[1-{\left(\frac{y}{R}\right)}^{2}\right]$$

The outlet pressures of the CoW model are modelled using Windkessel boundary conditions. The Windkessel boundary conditions are expressed through an ordinary differential equation *(ODE)* similar to the relation between voltage and current in electrical circuits. The ODE used for blood flow and particle simulations in the present work is shown in Equation 6, where *R_p_* is proximal resistance for large arteries and arterioles, *R_d_* is the distal resistant for simulating small arterioles and capillaries, *C* is vessel capacitance for large arteries and arterioles, *i*(*t*) represents the flow rate, and *p*(*t*) is the time dependent pressure.
(6)$$\left(1+\frac{R_{p}}{R_{d}}\right)i(t)+CR_{d}\frac{di(t)}{dt}=\frac{p(t)}{R_{d}}+C\frac{dp(t)}{dt}$$

The resistance and capacitance values used for modelling the Windkessel outlet boundary conditions are obtained from the work of Gharahi et al. [33]. Wall shear stress *(WSS)* and vorticity values are predicted on the fluid domain surfaces that represent the interface boundary between the fluid and the neighboring tissue. The equations used to calculate these hemodynamic parameters are shown below. Wall shear stress is calculated using Equation (7) and vorticity is calculated using Equation (8). Vorticity *ω* in Equation (8) is evaluated by ϖ≡∇×u, where *u* is the velocity.
(7)$$\tau_{yx}=\tau_{xy}=\mu \left(\frac{\partial u}{\partial y}+\frac{\partial v}{\partial x}\right)$$
(8)$$\frac{D\varpi }{Dt}=\left(\varpi \cdot \nabla \right)\overset{}{u}+v\nabla^{2}\varpi$$

### 3.2. Model Development

The CoW model evaluated in this work acts as the central blood distribution system in the brain and connects the inflow from the basilar and internal carotid arteries to the cerebral arteries via a circular system closed by communicating arteries. Studies have found considerable variation in the structure of this system. Its inherent redundancy allows it to function despite the presence of deformed or missing subsystems. Figure 14 depicts the modelling methodology of the CoW obtained from a CTA and produced in Solidworks^TM^ v22 and deployed in ANSYS software *(SpaceClaim and Meshing)*.

The process for generating this CAD model is similar to the methodology described in our previous works [13,26,34,35,36]. This process is outlined in detail in the schematic shown in Figure 14. Figure 15 shows the finalized CoW model with a table listing the names of the modeled arteries in the CAD model with the assigned boundary conditions.

### 3.3. Particle Force Balance

The physics describing the transport of magnetic nanoparticles in the cardiovascular system is governed by various factors: (1) the magnetic force produced by the external magnet, (2) viscous drag, (3) unsteady effects due to the driving pressure wave, (4) particle kinematics *(Brownian motion),* and (5) the interaction between the fluid and particles. In this work, our model incorporates the dominant magnetic and viscous forces and the particle/blood interaction through an integrated force balance. This is achieved via deriving the magnetic force exerted on a single particle which represents a cluster *(parcel)* of particles. Each particle has a radius *R_mp_* and a spherical volume *V_mp_*. The force exerted on the particles is derived from a dipole moment approach in which the particle is replaced by an equivalent point dipole which is focused on the center. The force computed on a parcel of particles in a non-conducting medium is described as:(9)$$F_{m}=\frac{4}{3}\pi R_{mp}{}^{3}\mu_{0}\frac{3\chi_{mp}}{\left(\chi_{mp}+3\right)}\left(H\cdot \nabla \right)H$$
where *H* is the applied magnetic field intensity at the center of the parcel, *χ_mp_* is the magnetic susceptibility of the magnetic particle, and *µ*_0_
*(4π × 10^−7^ N·A^−2^)* is the permeability in air [37]. The derivation of the magnetic force incorporates the assumption that blood is a non-magnetic medium with permeability. The magnetic field in this work is modelled as a cylindrical electromagnet of radius *R_mag_* that is positioned a distance *d* orthogonal to the axial flow field as shown in Figure 16. The magnetic field is described as:(10)$$H_{x}(x,y)=\frac{M_{s}R_{mag}{}^{2}}{2}\left(\frac{{\left(x+d\right)}^{2}-y^{2}}{{\left({\left(x+d\right)}^{2}+y^{2}\right)}^{2}}\right)$$
and
(11)$$H_{y}(x,y)=\frac{M_{s}R_{mag}{}^{2}}{2}\left(\frac{2(x+d)y}{{\left({\left(x+d\right)}^{2}+y^{2}\right)}^{2}}\right)$$

Equations (10) and (11) are substituted into Equation (9) to determine the magnetic force components as described below in Equations (12) and (13):(12)$$F_{mx}(x,y)=\frac{4}{3}\pi R_{mp}{}^{3}\mu_{0}\frac{3\chi_{mp}}{\left(\chi_{mp}+3\right)}\times \left[H_{x}(x,y)\frac{\partial H_{x}(x,y)}{\partial x}+H_{x}(x,y)\frac{\partial H_{x}(x,y)}{\partial y}\right]$$
and
(13)$$F_{my}(x,y)=\frac{4}{3}\pi R_{mp}{}^{3}\mu_{0}\frac{3\chi_{mp}}{\left(\chi_{mp}+3\right)}\times \left[H_{y}(x,y)\frac{\partial H_{y}(x,y)}{\partial x}+H_{y}(x,y)\frac{\partial H_{y}(x,y)}{\partial y}\right]$$

Equations (12) and (13) may be simplified via the differential product rule to yield the following magnetic field force components:(14)$$F_{mx}=\frac{-3\pi \mu_{0}\mu_{r}R_{mp}{}^{3}\chi_{mp}M_{s}{}^{2}R_{mag}{}^{4}}{\chi_{mp}+3}\frac{\left(x+d\right)}{2{\left({\left(x+d\right)}^{2}+y^{2}\right)}^{3}}$$
and
(15)$$F_{my}(x,y)=\frac{4}{3}\pi R_{mp}{}^{3}\mu_{0}\mu_{r}\frac{3\chi_{mp}}{\left(\chi_{mp}+3\right)}\times \left[H_{y}(x,y)\frac{\partial H_{y}(x,y)}{\partial x}+H_{y}(x,y)\frac{\partial H_{y}(x,y)}{\partial y}\right]$$

The derived model describes the targeting of a parcel of particles embedded with magnetite *(Fe_3_O_4_)* particles as described by Takayasu et al. [38] and Furlani and Furlani [37]. Magnetite nanoparticles are biocompatible and have a density of *ρ_p_* = 5230 kg/m^3^ [39]. For the magnetic field source, a 6 cm diameter *(radius 3 cm)* electromagnet, with a magnetization of *M_s_* = 1.5915 × 10^6^A/m *(remanence B_r_ = 2.0T*) is modelled. The surface of the magnet is positioned 7.8 cm from the centroid of the CoW as shown in Figure 16.

### 3.4. Particle Trajectory Model

A discrete phase model *(DPM)* was implemented in this work. This approach follows the Euler–Lagrange approach. The interaction with continuous phase option was enabled in Fluent to allow the discrete phase *(particles)* to exchange mass, momentum, and/or energy with the continuous phase *(blood).* The trajectory of the particles is predicted by integrating the force balance on the particle cluster, which is written in a Lagrangian reference frame. The force balance equates the particle cluster inertia with the forces acting on the particles and can be expressed as:(16)$$\frac{du_{p}}{dt}=F_{D}(u-u_{p})+\frac{g_{x}\left(\rho_{p}-\rho \right)}{\rho_{p}}+F_{bi}+F_{x}$$
where *u* is the fluid velocity, *u_p_* is the particle parcel velocity, *ρ* is the fluid density, *ρ_p_* is the particle density, *F_bi_* is the Brownian force acceleration term, *F_x_* is the body force acceleration term, *d_p_* is the particle diameter, and *F_D_*(*u − u_p_*) is the drag force per unit mass and is written as:(17)FD=18μρpdp2CDRe24

The relative Reynolds number, “Re” is defined as:(18)Re=ρdp|up−u|μ

The drag coefficient, “*C_D_*” is defined as:(19)CD=a1+a22Re+a33Re2
where *a*_1_, *a*_2_, and *a*_3_ are constants that apply for smooth particles using the spherical drag law and apply over a range of Reynolds numbers as reported by Morsi and Alexander [40]. For the present work, when substituting the appropriate constants, the drag coefficient takes the following form for micron sized particles:(20)CD=24Re(1+b1Reb2)+b3Reb4+Re
where
$$\begin{array}{c} b_{1}=\exp(2.3288-6.4581\phi +2.4486\phi^{2})\\ b_{2}=0.0964+0.5565\phi \\ b_{3}=\exp(4.905-13.8944\phi +18.4222\phi^{2}-10.2599\phi^{3})\\ b_{4}=\exp(1.4681+12.2584\phi -20.7322\phi^{2}+15.8855\phi^{3}) \end{array}$$
which is adapted from Haider and Levenspeil [41]. The shape factor, “*ϕ*” is defined as:(21)$$\phi =\frac{s}{S}$$
where *s* is the surface area of a sphere having the same volume as the particle, and *S* is the actual area of the particle. The shape factor cannot exceed a value of 1. For superparamagnetic particles, a form of Stokes drag law is used [42]. In this case, the drag force on a particle per unit mass is defined as:(22)$$F_{D}=\frac{18\mu }{d_{p}{}^{2}\rho_{p}C_{e}}$$
where the Cunningham correction factor *C_e_* is described as:(23)$$C_{e}=1+\frac{2\lambda }{d_{p}}\left(1.257+0.4e^{-(1.1d_{p}/2\lambda )}\right)$$
and *λ* is the molecular mean free path. For superparamagnetic particles, the effects of Brownian motion are included. The components of the Brownian force are modeled as a Gaussian white noise process with spectral density given by *S_n,ij_*.
(24)$$S_{n,ij}=S_{0}\delta_{ij}$$
where *δ_ij_* is the Kronecker delta function and:(25)$$S_{0}=\frac{216vk_{B}T}{\pi^{2}\rho d_{p}{}^{5}\left(\frac{\rho_{p}}{\rho }\right)C_{e}}$$

*T* is the absolute temperature of the fluid *(taken as body temperature), ν* is the kinematic viscosity, and *k_B_* is the Boltzmann constant. The Brownian force per unit mass component is described as:(26)$$F_{bi}=\zeta_{i}\sqrt{\frac{\pi S_{0}}{\Delta t}}$$
where *ζ_i_* is the zero-mean unit-variance-independent Gaussian random number.

### 3.5. Particle Capture Efficiency

One of the most reported challenges for MDT is the capability of targeting and capturing particles via producing a magnetic force to overcome the drag force acting on particles due to the inlet pressure wave. As a result, capture efficiency is a major parameter of interest in the present work. Capture efficiency describes the effectiveness of targeting particles under the influence of a magnetic field. The capture efficiency for the magnetized section of the artery is defined as the ratio of the number of injected particle parcels to the number of particle parcels leaving the magnetized region:(27)$$\eta_{c}=\frac{N_{np,in}}{N_{np,out}}\times 100\%$$

In this work, Equation (27) is evaluated via imposing a reflective boundary condition at the wall of the magnetized section *(region of interest (ROI)).* Previous studies have utilized a trapped boundary condition as opposed to a reflective boundary condition [43]. A trapped boundary condition cancels the trajectory mapping of particle trajectories as opposed to continuously monitoring the trajectory if the particles move from the current position in a future time step. Additionally, an escape boundary condition is imposed at the outlets which allowed particles to escape *(exit)* the outlets. The capture efficiency is determined via comparing the number of injected particle parcels *N_np,in_* to the number of escaped particle parcels *N_np,out_* as described in Equation (27). The next section describes the characteristic non-dimensional parameters evaluated in this work for characterizing dominant effects.

### 3.6. Characteristic Non-Dimensional Parameters

The magnetic field strength settings for this work are 0, 2, 4, 6, and 8T which are within the FDA suggested range [44]. Nanoparticles larger than 200 nm are typically filtered by liver and spleen while those smaller than 10 nm are easily cleared by the kidney or through extravasations during the blood circulation [45,46]. For this work, we are evaluating particles ranging from 10 nm to 4 µm. Two dimensionless numbers, *β_m_* and *Pe_m_*, are evaluated relative to the magnetic field, magnetic force induced particle motion, and convective blood flow. *β_m_* measures the ratio of the time for particles to reach the wall by diffusion to that by magnetic force as shown below:(28)$$\beta_{m}=\frac{d_{c}^{2}/6D}{d_{c}/u_{m}}=\frac{d_{c}u_{m}}{6D}$$
where *d_c_* is the capture distance, which is chosen to be 50 µm as mentioned above, *D* is the diffusion coefficient, and *u_m_* is the magnetic field-induced velocity in the normal direction of the stent surface. The diffusion coefficient can be calculated by the following expression:(29)$$D=\frac{kT}{6\pi \mu r}$$
where *k* and *T* are again the Boltzmann constant and temperature, respectively. If *β_m_* >> 1, this indicates that the capture process of particles is magnetic force dominated. If *β_m_* << 1, this indicates that the capture process of the particles is diffusion dominated. The modified Peclet number for this work describes the ratio between particle radial traveling time toward the wall and the convection time in the channel. This number provides insight on whether the flushing effect or capturing effect is dominant in the particle delivery as shown below:(30)Pem=dc2μReρdL(6D+umdc)

After release, the particles are transported in the channel under effects of external magnetic field, high gradient magnetic field induced by the stent, flow drag force, and BM. When distances between the particles and the artery wall are smaller than the particle radius, they are treated to be captured by the stented region.

### 3.7. Numerical Methods 

The pulsatile flow of blood and particle tracking was calculated via ANSYS Fluent^TM^ 22. The magnitude of the magnetic force components is substituted into the accelerated body force term in Equation (16). User defined functions were written for the pulsatile velocity waveform and the magnetic force generated by the magnetic field. The user defined functions were compiled in the ANSYS Fluent platform for simultaneous flow and particle trajectory calculations. The Navier Stokes equations were solved implicitly using a quadratic upwind discretization scheme *(QUICK)* for nonlinear terms. The integrated force balance described in Equation (16) was numerically integrated using a sixth order Runge–Kutta scheme for instances where the left side of the equation was significant, and a Euler scheme otherwise. A two-way fluid-particle coupling method is implemented in Fluent to predict the effect of the discrete phase on the continuum. This method solves the discrete and continuous phase equations until the solutions of both phases have stopped changing.

The flow domain was discretized into a large number of tetrahedral computational cells with mesh inflation. The arterial model was tested for four different mesh grid densities, i.e., 728, 932, 1096, and 1385 cells in the cross-sectional flow area. The time-averaged absolute difference in the basilar inlet centerline axial velocity between the coarse and fine cross-sectional mesh was 1.2 mm/s, and that between fine and finer one was only 0.07 mm/s. In addition, when comparing velocity and maximum wall shear stress the chosen grid resulted in values of less than 5% difference from the previous grid *(finest grid).* Mesh inflations were also used in areas where appropriate do reduce mesh irregularities. Figure 17 shows the coarsest mesh from the grid independent study with the domains.

## 4. Conclusions

The computational MDT simulations of particle flow and targeting in the CoW model presented in this work demonstrate that the greatest particle capture efficiency results for particle diameters within the micron range, specifically 0.7 to 4 µm in regions where flow separation and vortices are at a minimum. For smaller particles ranging from the superparamagnetic regime to 100 nm, capture efficiencies of 20–99% were observed. This is an improvement over our previous work in which we observed a capture efficiency range of 0.5–30% for particles ranging in diameters of 20 to 400 nm and was not subjected to a magnetized implant *(as reported in our previous work).* Capture efficiencies in the superparamagnetic regime have tripled compared to our previous work when considering a CoW model and varying the field strength (*2T–8T*). Similar to other works investigating MDT in larger arteries, it was also determined that the capture efficiency of particles decreases with particle diameter similar to our previous work. Contrary to previous reported works, the use of a magnet positioned from the centroid of the CoW model increased the particle capture efficiency for superparamagnetic particles. The key finding in this work is that favorable capture efficiencies for superparamagnetic particles were observed in the CoW model for weak fields (*B_r_ < 4T*). Although capture efficiency in this work has been improved, several simplifications have been introduced in the computational simulations reported in this work.

Some simplifications in this work involved simplifying the vessel geometry, boundary conditions, and particle modelling. These simplifications include restructuring portions of the arterial model that promote meshing irregularities, modelling the vessel as a rigid wall vessel, implementing a velocity inlet waveform that does not match the true waveform of the subject being studied, and not modelling the drug thickness on carrier particles. From a geometry perspective, the CoW has a highly diverse structure, whereas the CoW presented in this work appears in 50% or less of the world population [47]. CoW vessel geometry will influence the hemodynamic features and the hydrodynamics of MDT carrier particles and future work should consider case studies of different CoW patient-specific geometries to gain perspective of the effect of vessel geometry on particle targeting efficiency. 

An additional significant simplification was made by evaluating the drag force for superparamagnetic particles via Stokes expression, using the Carreau non-Newtonian fluid viscosity model. The Carreau fluid viscosity model projects a significantly higher viscosity than the actual viscosity of blood plasma. In this case, small particles within the superparamagnetic regime may tend to move along stream traces in the blood plasma without colliding with blood constituents, therefore experiencing a drag force proportional to the viscosity of the blood plasma.

The viscous drag experienced by particles of the order of or smaller than the blood cells, as used in the present work, may therefore be much smaller than that which is presently used by considering the Carreau viscosity model. This could significantly change the outcome of capture efficiency for superparamagnetic particles in computational simulations such as those presented in the present work. The obtained capture efficiency results for the superparamagnetic particles could be significantly underestimated as mentioned in our previous work. Computational simulations, such as those discussed in the present work, make it possible to study the feasibility and practicality of MDT a prior to clinical trials. Furthermore, computational simulations are useful for investigating the influence of various factors independently and for optimization. The simulation results presented in the present work yield favorable capture efficiencies for micron range particles and superparamagnetic particles in smaller arteries using magnetized implants such as stents. The present work presents results for justifying further investigation of MDT. Further work should include an experimental investigation of MDT in mock arterial vessels to validate and verify the results presented in this work.

## Figures and Tables

**Figure 1 ijms-24-02545-f001:**
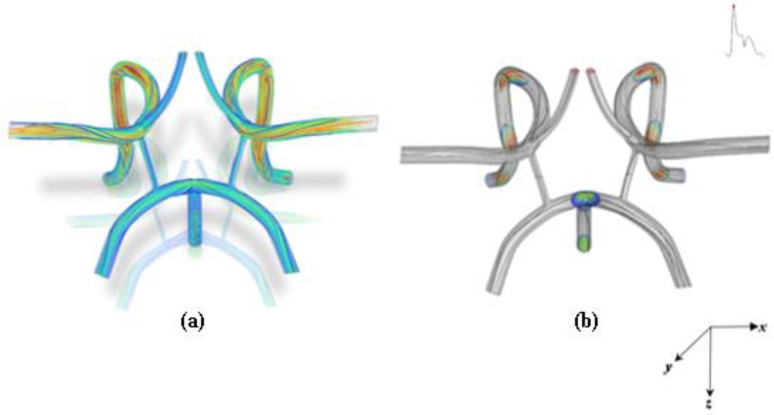
Flow characteristic profile inside the CoW: (**a**) Streamline traces of blood flow and (**b**) plot segments of velocity contour and streamlines during systole *(5th cardiac cycle)*.

**Figure 2 ijms-24-02545-f002:**
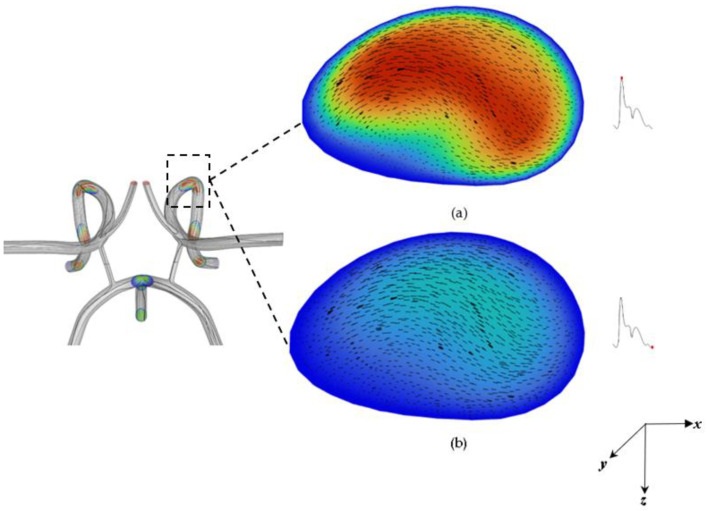
Cross-sectional view of the velocity contours and vector at the middle cerebral artery segment at: (**a**) systole and (**b**) diastole.

**Figure 3 ijms-24-02545-f003:**
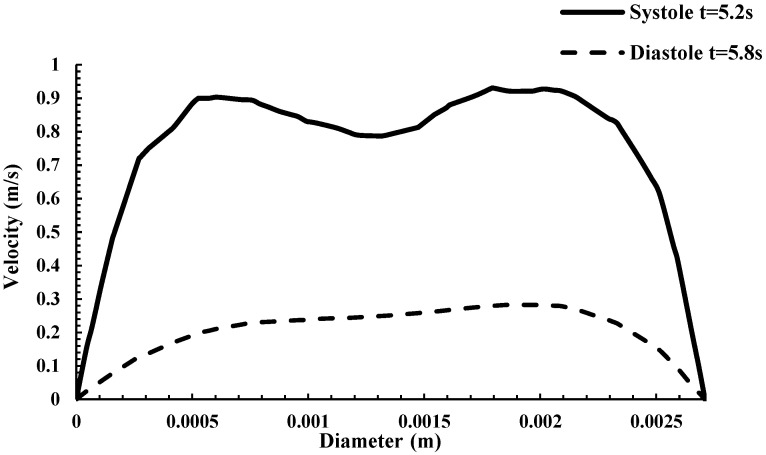
Velocity profiles at the cerebral artery branch contour segment at systole and diastole.

**Figure 4 ijms-24-02545-f004:**
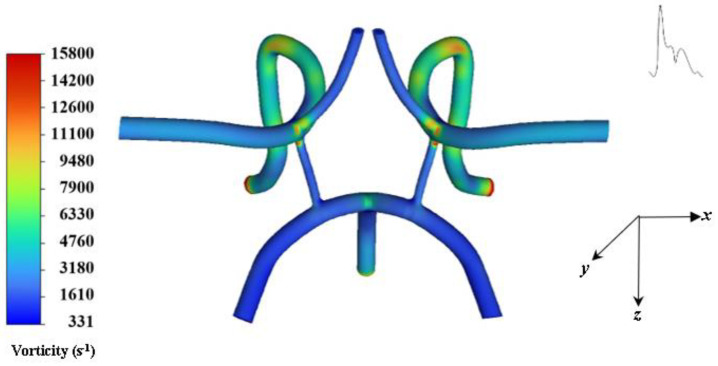
Contour of flow vorticity at systole.

**Figure 5 ijms-24-02545-f005:**
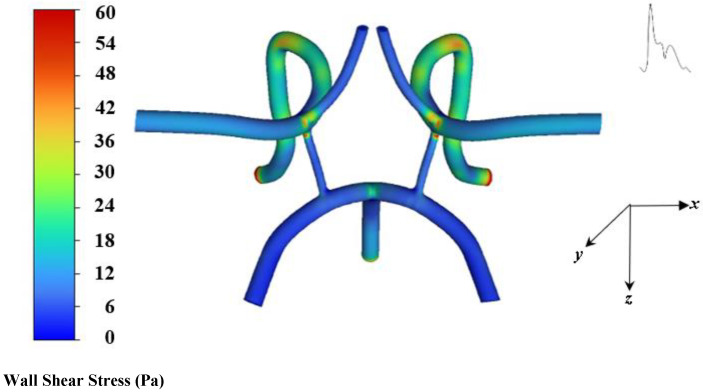
Contour of wall shear stress at systole.

**Figure 6 ijms-24-02545-f006:**
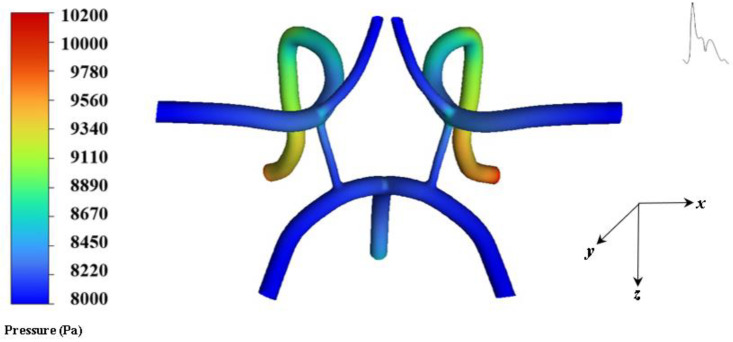
Contour of pressure at systole.

**Figure 7 ijms-24-02545-f007:**
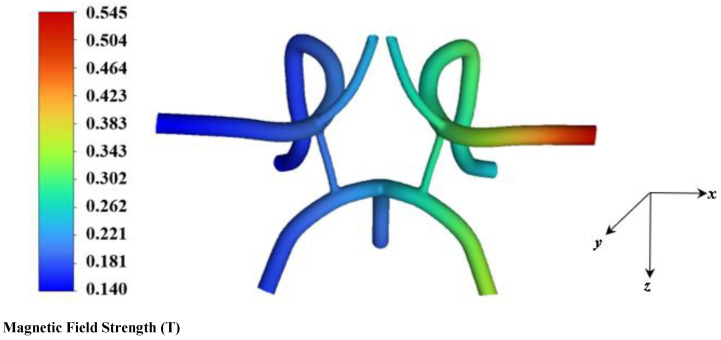
Contour of the arterial magnetic field strength at *B_r_* = *2.0T* over the entire fluid arterial domain generated in ANSYS Fluent.

**Figure 8 ijms-24-02545-f008:**
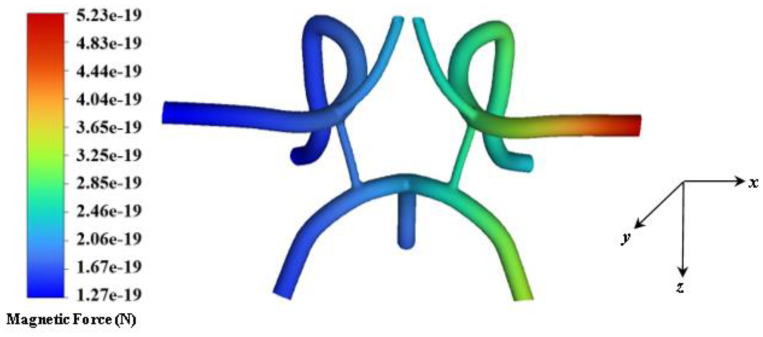
Contour of force magnitudes as a response to the magnetic field strength *(at B_r_ = 2.0T)* over the entire fluid arterial domain generated in ANSYS Fluent.

**Figure 9 ijms-24-02545-f009:**
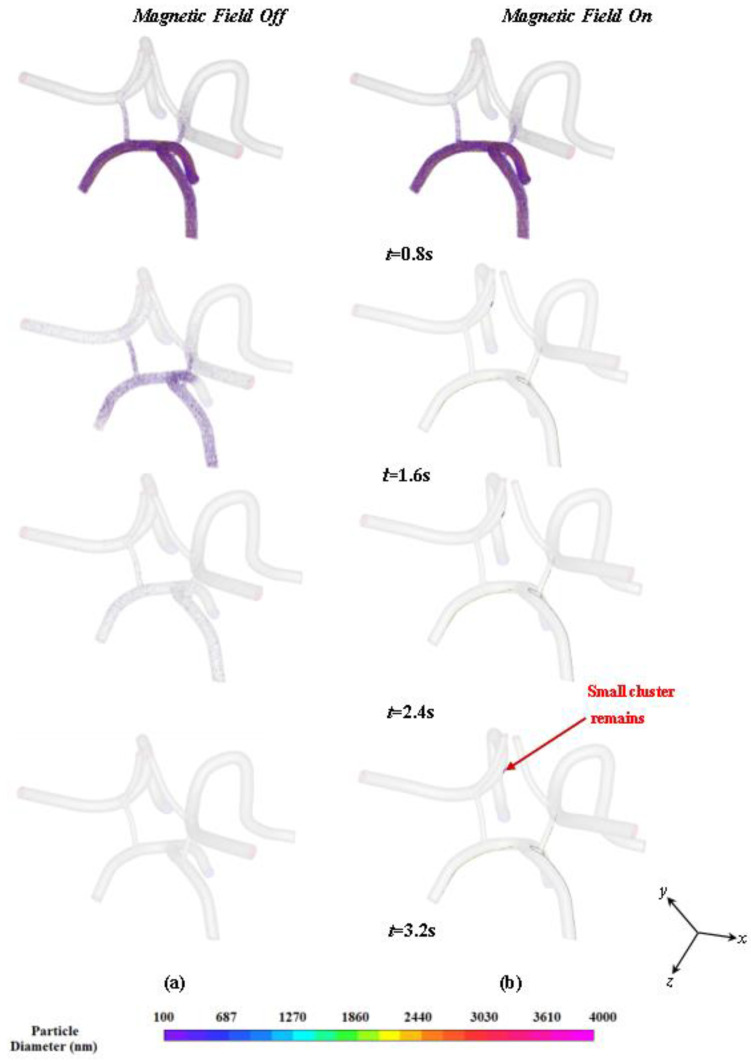
Heterogenous particle distribution in the CoW model at different time instances: (**a**) without a magnetic field and (**b**) with a magnetic field turned on at 0.8 s and for the duration of the tracking.

**Figure 10 ijms-24-02545-f010:**
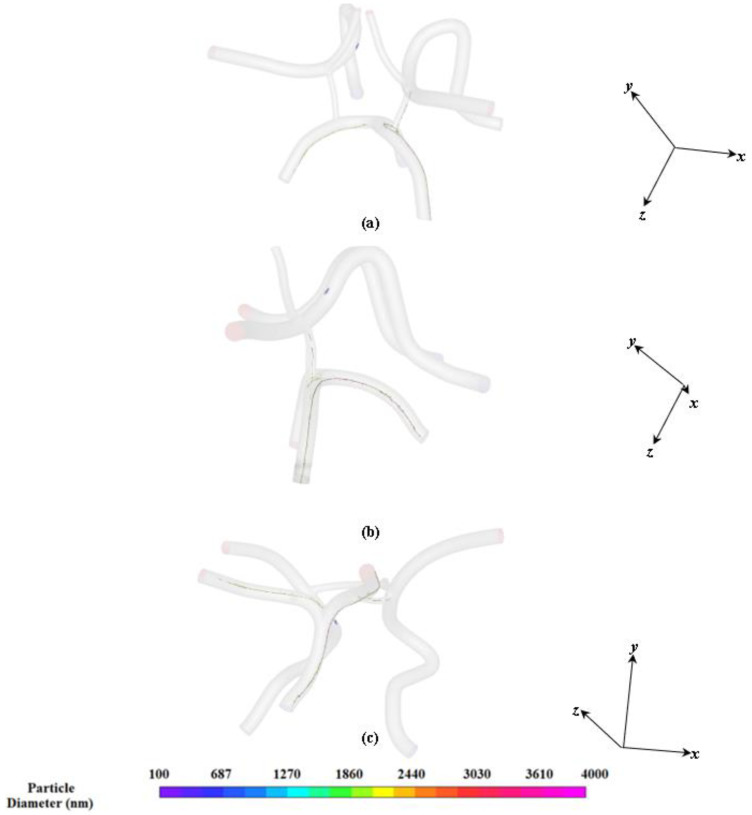
Heterogeneous particle distribution at different views: (**a**) isometric view, (**b**) side view, and (**c**) bottom view.

**Figure 11 ijms-24-02545-f011:**
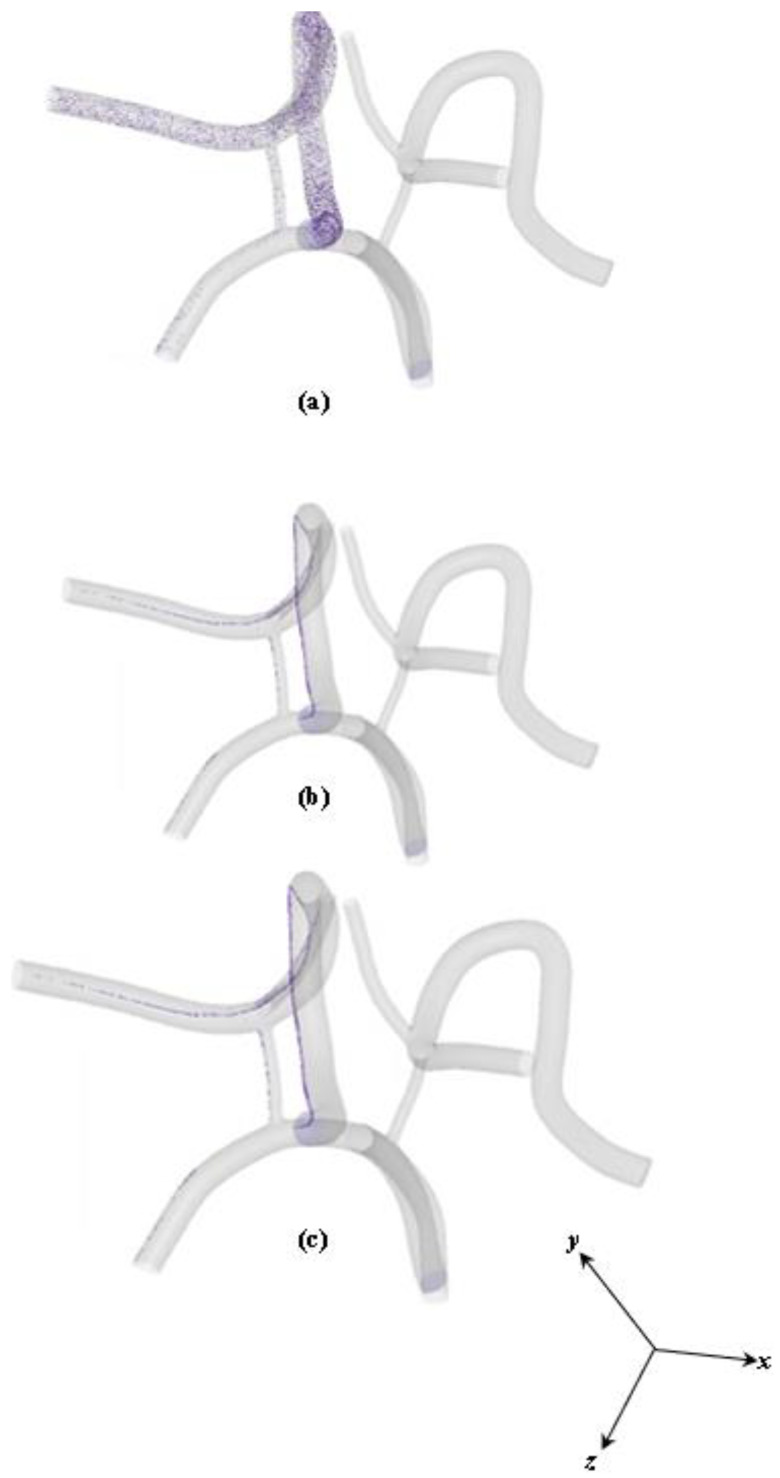
Homogeneous particle diameter of 10 nm distribution at different views: (**a**) isometric view, (**b**) side view, and (**c**) bottom view.

**Figure 12 ijms-24-02545-f012:**
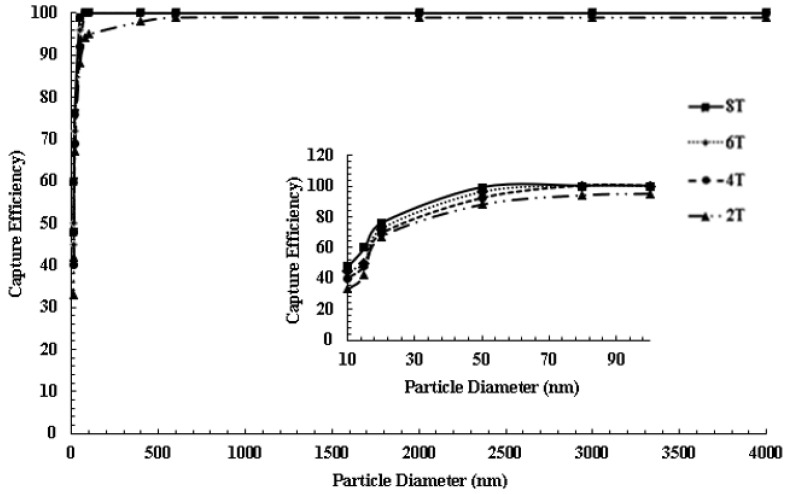
Plot of capture efficiency as a function of particle diameter in the simulated arterial flow. The inset shows the efficiency of MDT in the superparamagnetic regime.

**Figure 13 ijms-24-02545-f013:**
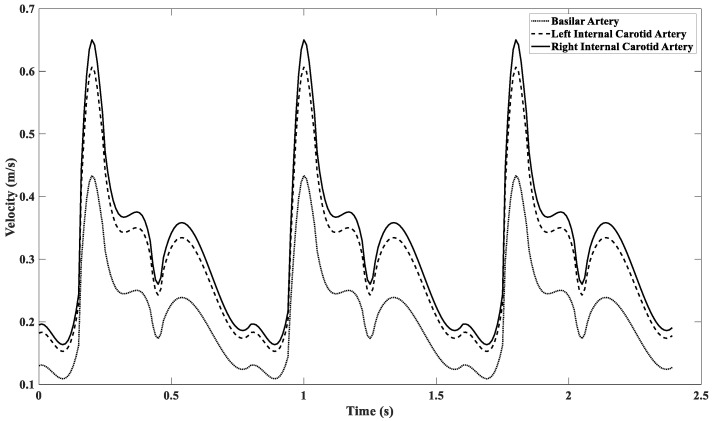
Plot of unsteady inlet velocity waveforms.

**Figure 14 ijms-24-02545-f014:**
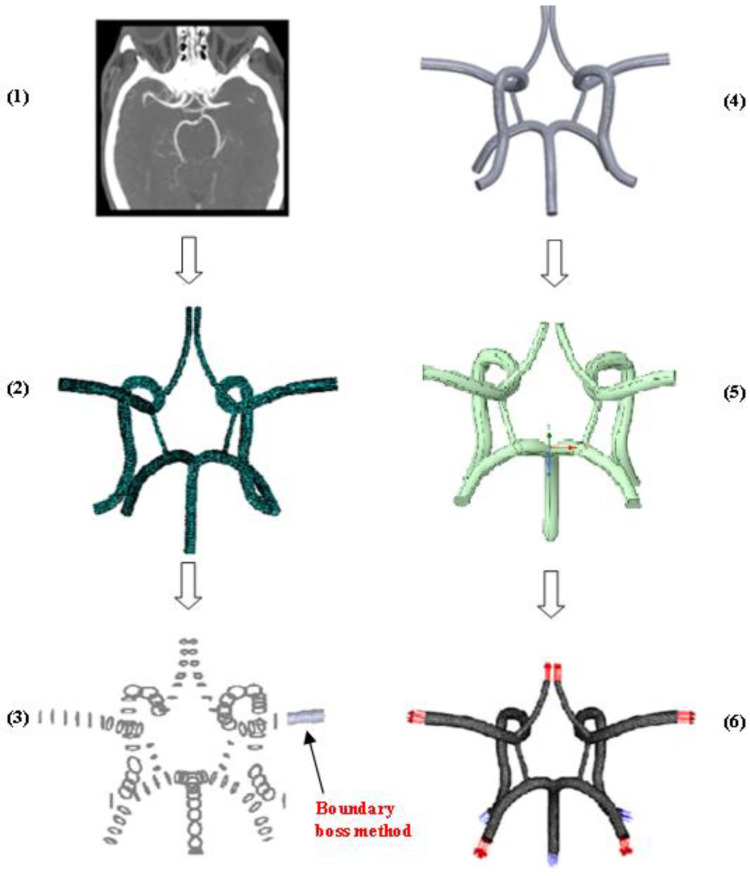
CoW computational model development methodology: (1) CTA scan, (2) point cloud image deployment, (3) cross sectional sketch uniting via boundary boss bass method, (4) full fluid core model after uniting the sketches, (5) Ansys SpaceClaim face healing, and (6) the final mesh and boundary condition assignment in Ansys Meshing.

**Figure 15 ijms-24-02545-f015:**
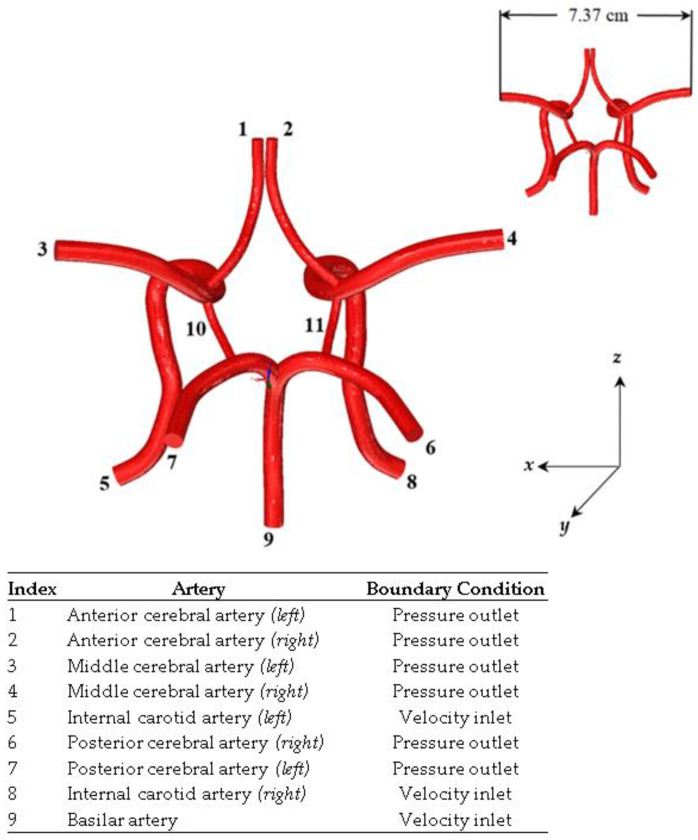
CAD fluid volume model of the Circle of Willis and the numbering of the inlet/outlet. This figure is cross-referenced with a table.

**Figure 16 ijms-24-02545-f016:**
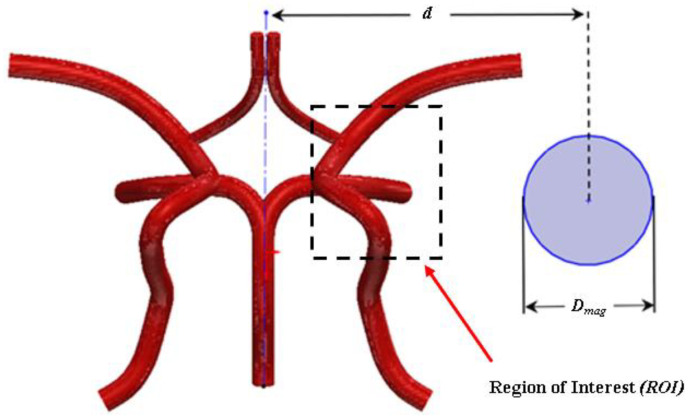
Schematic representation of the circular electromagnet of radius “*R_mag_*” distanced “*d*” from the centroid of the CoW arterial model and the ROI site.

**Figure 17 ijms-24-02545-f017:**
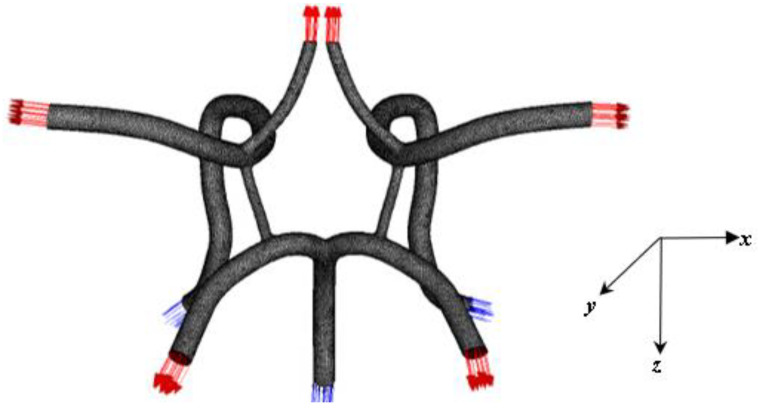
Computational CoW arterial mesh *(coarsest mesh)*.

## Data Availability

Data may be available upon request to the corresponding author.

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
