# Peer review of "Computational Assessment of Magnetic Nanoparticle Targeting Efficiency in a Simplified Circle of Willis Arterial Model"

_ijms, 2023, doi:10.3390/ijms24032545_

Round 1
Reviewer 1 Report
In this work, the authors performed numerical simulations of magnetic nanoparticle targeting efficiency in a Circle of Willis (CoW) model. The obtained results and the methodology presented in this work is helpful to assess the feasibility and practicality of medical drug targeting. The paper is well organized, technically sounds.
I think this paper is accepted after some minor revisions.
Questions and comments:
1. How did the authors calculated the magnetic susceptibility of the magnetic particles in equation (9)? In general, the magnetic susceptibility of the magnetic particles exhibits nonlinear magnetization properties.
2. What does mu_r mean in equation (9)?
3. (page 6, line 150) The number of the equation is not (3) but (5).
4. Some figure numbers in the main text are wrong.
(page 5, line 174), (page 7, line 226), (page 14, line 440), (page 16, line 487), (page 16, line 491)
Author Response
Questions and comments:
- How did the authors calculated the magnetic susceptibility of the magnetic particles in equation (9)? In general, the magnetic susceptibility of the magnetic particles exhibits nonlinear magnetization properties.
Thank you for pointing this out. The susceptibility was calculated based on the work cited in previous work as documented.
- What does mu_r mean in equation (9)?
Thank you for pointing this out. This has been corrected. The mu_r should not be in any equations.
- (page 6, line 150) The number of the equation is not (3) but (5).
Thank you for pointing this out. This has been corrected
- Some figure numbers in the main text are wrong.
(page 5, line 174), (page 7, line 226), (page 14, line 440), (page 16, line 487), (page 16, line 491)
Thank you for pointing this out. This has been corrected
Reviewer 2 Report
In this study, the authors explored the MDT in CoW using computational modelling. Overall, it is well structured with a logic flow. However, some details need further improvement to achieve publishable standards.
1. In the mesh independence study, please provide the relative changes in percentage.
2. Did you perform any refinement of mesh at certain locations. e.g. near-wall areas?
3. Regarding the inlet boundary conditions, are they synchronized? There would be some minor delay, although difficult to show in the figure but may cause hemodynamic changes in the arteries.
4. What are the mechanical models for artery wall and intracranial tissues?
5. Please double check Figure 17. By the way, I would like to suggest some unnecessary figures and make the total number of figures within 10.
6. The CoW model is an idealized one but an radiological image was shown in figure 2. Please provide the details of geometric reconstruction.
7. Please separate the discussion and conclusion sections. The conclusion should be brief, preferably within 3-5 lines.
8. It need to be mentioned in the discussion that CoW has a highly diverse structure, where the intact CoW appears in only 50% of the population, which will influence the hemodynamic features and has been considered in some recent computational models (Refer: 10.1109/ACCESS.2020.3007737). It is preferable to simulate different CoW strutures, or at least to discuss this as a limitation, and consider it in future studies.
Author Response
Reviewer 2
- In the mesh independence study, please provide the relative changes in percentage.
Thank you, this is mentioned on page 11.
- Did you perform any refinement of mesh at certain locations. e.g. near-wall areas?
Thank you, this is mentioned on page 11.
- Regarding the inlet boundary conditions, are they synchronized? There would be some minor delay, although difficult to show in the figure but may cause hemodynamic changes in the arteries.
Thank you for your suggestion. The boundary conditions are synchronized and the limitation is discussed in the conclusion.
- What are the mechanical models for artery wall and intracranial tissues?
Thank you for your question. As mentioned on page 3, The walls of the arterial vessels within the CoW are modeled as rigid vessels with the no-slip condition applied at the boundaries. The model does not expand and contract and this is also mentioned in the limitations.
- Please double check Figure 17. By the way, I would like to suggest some unnecessary figures and make the total number of figures within 10.
Thank you for your suggestion. We have double checked the figure and would like to include all figures. In the past when we haven’t included such figures, reviewers and even authors have asked where are these figures. We believe this figures will provide a full picture.
- The CoW model is an idealized one but an radiological image was shown in figure 2. Please provide the details of geometric reconstruction.
We mentioned that the CoW was generated form a magnetic resonance imaging (MRI) scan and that the computer aided design model (CAD) of the model was produced in SolidworksTM v22 “The process for generating this CAD model is similar to the methodology described in our previous works [13, 26, 29-31]”
- Please separate the discussion and conclusion sections. The conclusion should be brief, preferably within 3-5 lines.
Thank you for your suggestion. We’ve separated the sections but also have included in the conclusion the case points/limitations you mentioned earlier that may effect targeting efficiency in which future researchers should consider.
- It need to be mentioned in the discussion that CoW has a highly diverse structure, where the intact CoW appears in only 50% of the population, which will influence the hemodynamic features and has been considered in some recent computational models (Refer: 10.1109/ACCESS.2020.3007737). It is preferable to simulate different CoW strutures, or at least to discuss this as a limitation, and consider it in future studies.
Thank you for your suggestion, the conclusion is revised to include this.
Round 2
Reviewer 2 Report
Thanks for the update. Some points still need major revision/improvement for further clarification.
1. Regarding the boundary conditions, in Equation 4, do you mean that Ci is multiplied to the items? If so, basilar and carotid arteries have proportional flow velocity, which does not conform to the physiological condition where the flow in carotid and basilar arteries differ in both phase and waveform. Please explain.
2. The artery geometry reconstruction need more details. In the revised version it was mentioned that "arterial vessels have been simplified to a constant diameter along the center axis", which is confusing since there might be unevenness at the connection. Instead of citing your existing works where the methods are different from the current one, please provide a supplement instead to cover the technical details of geometric reconstruction with essential illustration.
3. Since you are calculating WSS, near-wall mesh refinement should be considered.
4. Regarding the anatomic diversity "CoW pre- 693 sented in this work appears in 50% or less of the world population", please cite the reference properly.
Author Response
We appreciate your concerns and have addressed them.
- Regarding the boundary conditions, in Equation 4, do you mean that Ci is multiplied to the items? If so, basilar and carotid arteries have proportional flow velocity, which does not conform to the physiological condition where the flow in carotid and basilar arteries differ in both phase and waveform. Please explain.
Yes, that is what the equation displays. To clarify further, we’ve added a multiplication/product sign to indicate that the constant is multiplied. We also mentioned several times that the waveforms are simplified. We also mentioned before in the paper in paraphrased form in the conclusion that the simplified waveform may have an effect on the hemodynamics.
“implementing a velocity inlet waveform that does not match the true waveform of the subject being studied” (listed in the conclusion)
- The artery geometry reconstruction need more details. In the revised version it was mentioned that "arterial vessels have been simplified to a constant diameter along the center axis", which is confusing since there might be unevenness at the connection. Instead of citing your existing works where the methods are different from the current one, please provide a supplement instead to cover the technical details of geometric reconstruction with essential illustration.
The goal of the paper is not the construction of the arterial geometry, but the hemodynamics as a result of the geometry. Our previous paper lists this methodology as well as other works which goes into detail of the model construction. This would lengthen the paper to include this, when in your first comments you mentioned there were too many figures and that the conclusion needed to be reduced to three lines which is very uncommon.
Additionally, there are a lot many works that do not focus on this or go into more detail other than mentioning short details of the construction.
- Since you are calculating WSS, near-wall mesh refinement should be considered.
We also mentioned that inflation was performed in areas where appropriate.
- Regarding the anatomic diversity "CoW pre- 693 sented in this work appears in 50% or less of the world population", please cite the reference properly.
Thank you for pointing this out, we have now addressed this with a citation from your previous works. We’ve included your citation.
Round 3
Reviewer 2 Report
Thanks for the update. My major concerns still lies in the second point, i.e., geometry reconstruction. Since the journal permits to update the supplementary materials, I recommend you provide the details in a supplement.
Author Response
Thank you for your comments. To make things much easy and simpler, we have revised figure 2 and the paragraph before that. Figure 2 now includes a schematic that shows the processing methodology. As depicted in Figure 2, the computer aided design (CAD) volume rendering methodology of the CoW is obtained from a CTA and produced in SolidworksTM v22 and deployed in ANSYS software (Space Claim and Meshing). The process for generating this CAD model is similar to the methodology described in our previous works [13, 26, 29-31]. This process involves deploying the rough mesh file generated from the CTA point cloud data file, creating planes, slices, and sketches along the axis of each artery and uniting the sketches using the loft boundary boss feature in Solidworks. Fillets are applied in sharp regions where the arteries bifurcate/transition. Further post processing is done in ANSYS Space Claim such as healing faces and transitions. The model is finally meshed in ANSYS meshing. This is included in the new paragraph and we mentioned that Section 2.7 provides details on the meshing methodology. We hope that this answers your concerns.